# Exploratory Analysis of Scholarly Publications on Artificial Intelligence (AI) in Colonoscopy using Litstudy

**Mary Adewunmi & Kwang Chien Yee**
CHM, School of Medicine,
University of Tasmania(UTAS),
Hobart, TAS7000, Australia.
{mary,adewunmi}{kwang,yee}@utas.edu.au

**Dr Ming Chao Wong**
School of ICT,
University of Tasmania(UTAS),
Sandybay, TAS7005, Australia.
{ming,wong}@utas.edu.au

## Abstract

Due to the large number of scholarly papers on AI and colonoscopy and the short research period, it can be difficult to answer general questions about the research area, such as who the key authors are and what the key issues or insights are. We use Litstudy, a Python library, to study colonoscopy AI research. "AI" and "colonoscopy" keywords were used as search results. 3865 IEEE Xplore and 2007 Springer bibliographies were downloaded. Scopus found 5083 citation papers, excluding 789 unavailable citations. Topic clusters were created using the NMF model with a 0.85 threshold. Topic clouds showed that "Patient" occurred most in four topics: 2, 3, 7, and 10. Despite querying IEEE, Springer, and Scopus databases with the "Artificial Intelligence" keyword, subject 5 with AI has the lowest topic phrase weight in topic clouds. Topic 10 words cluster on colon cancer rehabilitation in colonoscopy showed weak topic clusters. The project selects scientific articles, analyses and visualises their scholarly contribution using natural language processing (NLP), bibliographic network analysis, and, most importantly, reveals word clusters in AI for colonoscopy publications.

## 1 Introduction

AI systems aid colonoscopyXu et al. (2023)Mori et al. (2021). Litstudy Heldens et al. (2022) [1], for the selection of scholarly scopus articles, visualisation with bibliographic network analysis that show connections between publications and their authors, graph statistics on document metadata and automated topic discovery using natural language processing.

## 2 Methodology

### 2.1 About the datasets

"AI" AND "Colonoscopy"[2] keywords were used as search results for primary metadata which includes the title, author, publication date, DOI (Digital Object Identifier) and the secondary data generated were topic clusters from abstract and bibliographic cocitation networks from many references/citations extracted. 3865 papers from IEEE Xplore and 2007 papers from Springer making 5872 papers in CSV format. Out of which 5083 citation papers were found on Scopus and 789 unavailable citations were discarded.

### 2.2 Followed Methods

We used the litstudy libraryHeldens et al. (2022) in this project, installed through the Python Package Index (PyPi) Bommarito & Bommarito (2019). It is based on several well-known tools from the

---

[1]Litstudy is a Python module that makes it possible to respond to queries using simple scripts

[2]Colonoscopy is the standard method of removing polyps in Colon cancer patientsRex et al. (2015)

Python data science ecosystem, including Pandas McKinney et al. (2011); McKinney (2012) and NumPyOliphant et al. (2006). Furthermore, we used non-negative matrix factorization (NMF) an unsupervised model for topic detection and semantic relationship using a threshold of 0.85 [4] for the simplicity and easier interpretability of its results. The five key steps are as follows: Explored and downloaded scientific document metadata from a variety of sources, including IEEE, Springer's multiple research databases. The bibliography data from several sources was merged into a comma-separated variable (CSV) format; Sorted, chose, removed duplicates from, and annotate document collections by comparing them with the Scopus bibliography data; Calculated and displayed broad statistics about the documents' metadata (e.g., statistics per year, per author, per journal, etc.); Created, mapped, and evaluated different bibliographic networks that illustrate the connections between the use of AI for colonoscopy publications and their authors, and topic discovery automatically using natural language processing (NLP).

## 3 RESULTS AND DISCUSSION

According to Fig.1, patients was highly prevalent in four topics: 2, 3, 7, and 10, which we further validated using cocitation graph as seen in Fig.2. which was similarly used by Song et al. (2023) to study relationship among authors in scientometrics. Despite utilising the same keyword to search the database, subject 5 with AI has the lowest topic phrase weight.

Topic 10 in Fig.1 on cancer rehabilitation [3] revealed a weak relationship among the topic clusters, which requires additional investigation with the foreknowledge that the subject has a significant impact on colon

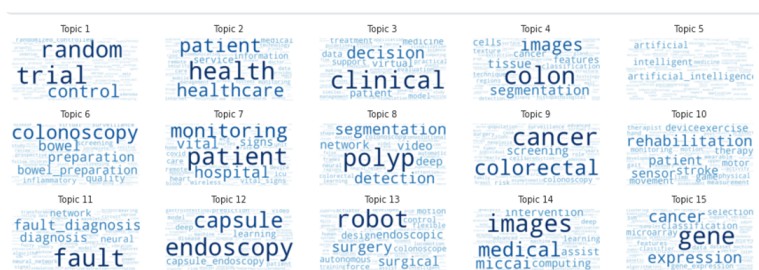

Figure 1: Top Word Clusters

cancer recovery.

In retrospect, co-citation analysis evaluates AI and Colonoscopy document similarities. Future co-citation frequencies fluctuate with academic field evolution.

## 4 CONCLUSION

We examined AI in colonoscopy research using Litstudy, a Python application. This lets us apply natural language processing, topic matrix model, and cocitation network analysis to analyse research publications on colonoscopy and artificial intelligence. The study also revealed that artificial intelligence has been understudied in colonoscopy and colon cancer in general, and some topics or subjects require more attention than others.

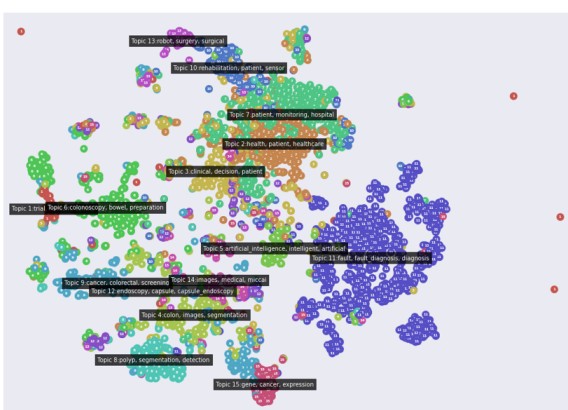

Figure 2: Cocitation Bibliography Network Graph

---

[3]Rehabilitation is one of the procedures used to follow up on cancer recovery. van Weert et al. (2005)

### ACKNOWLEDGEMENTS

Special thanks to Elsevier customer support, Paul Jordan Biyo, and UTAS librarian, Nathaniel Enright, for helping out with sorting Scopus API errors during the coding phase.

### URM STATEMENT

Author MA meets the URM criteria of ICLR 2023 Tiny Papers Track.

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

## A  APPENDIX

### A.1  CODES

The Notebook and the data would be added after the paper must have been accepted.

### A.2  OTHER FIGURES FROM RESULTS

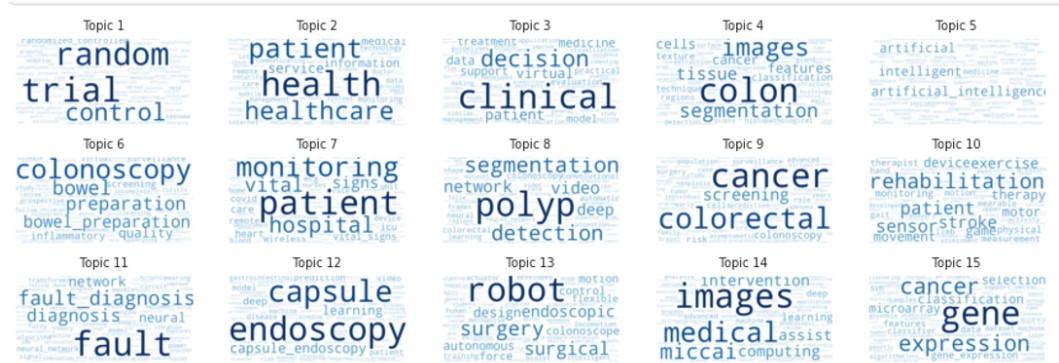

Figure 3: Top Word Clusters

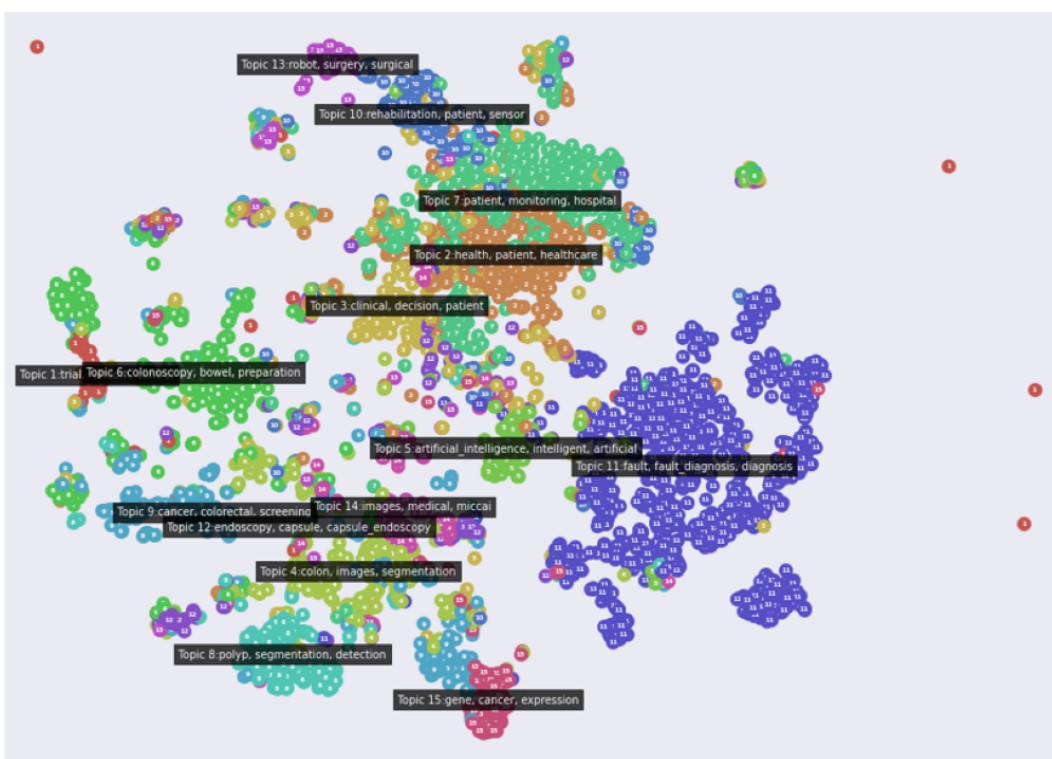

Figure 4: Cocitation Bibliography Network Graph

```
Topic 1: ['trial', 'random', 'control', 'randomized_controlled', 'adenoma']
Topic 2: ['health', 'patient', 'healthcare', 'service', 'information']
Topic 3: ['clinical', 'decision', 'patient', 'virtual', 'data']
Topic 4: ['colon', 'images', 'segmentation', 'tissue', 'features']
Topic 5: ['artificial_intelligence', 'intelligent', 'artificial', 'medicine', 'application']
Topic 6: ['colonoscopy', 'bowel', 'preparation', 'bowel_preparation', 'quality']
Topic 7: ['patient', 'monitoring', 'hospital', 'vital', 'signs']
Topic 8: ['polyp', 'segmentation', 'detection', 'video', 'network']
Topic 9: ['cancer', 'colorectal', 'screening', 'colonoscopy', 'risk']
Topic 10: ['rehabilitation', 'patient', 'sensor', 'stroke', 'exercise']
Topic 11: ['fault', 'fault_diagnosis', 'diagnosis', 'network', 'neural']
Topic 12: ['endoscopy', 'capsule', 'capsule_endoscopy', 'learning', 'deep']
Topic 13: ['robot', 'surgery', 'surgical', 'endoscopic', 'design']
Topic 14: ['images', 'medical', 'miccai', 'intervention', 'computing']
Topic 15: ['gene', 'cancer', 'expression', 'classification', 'microarray']
```

Figure 5: Topwords with NNMF model

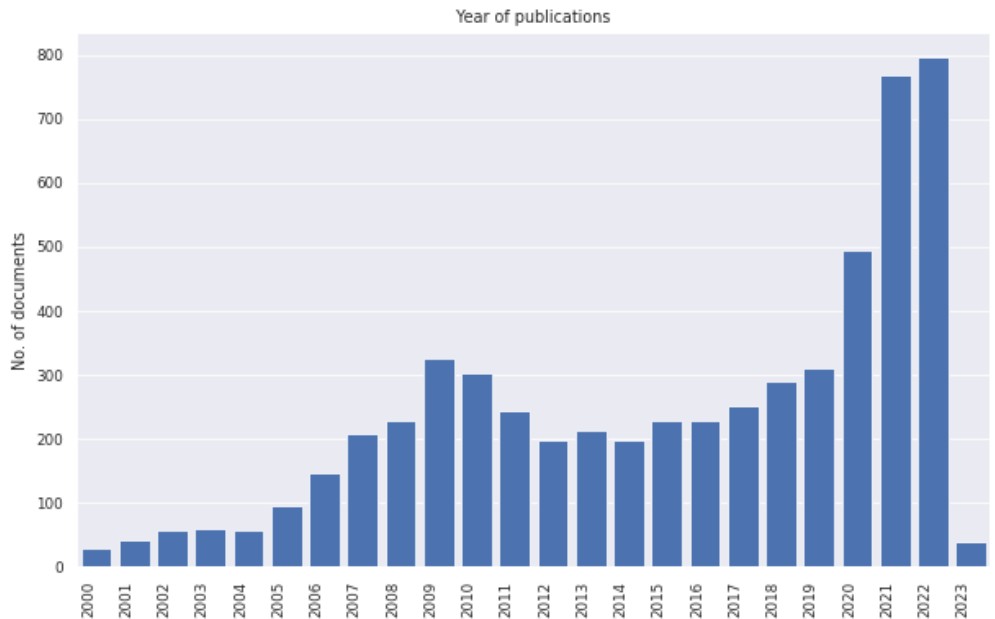

Figure 6: Year of Publication Histogram

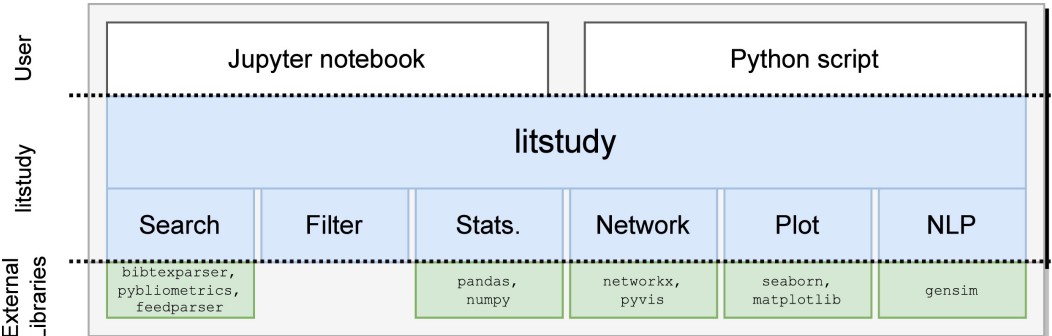

Figure 7: Litstudy conceptual frameworkHeldens et al. (2022)

