# OpenReview forum: "Exploratory Analysis of Scholarly Publications on Artificial Intelligence (AI) in Colonoscopy using Litstudy"
_ICLR.cc/2023/TinyPapers — Submitted to Tiny Papers @ ICLR 2023_

### Official Review · Reviewer_VguM · 2023-03-31

**Confidence:** 5

**Summary Of Contributions:**

The paper presents an exploratory analysis of scholarly publications on artificial intelligence (AI) for colonoscopy. Paper utilizes the Litstudy python library to select articles and automatically generate co-citation network analysis showing connections between authors, and article topics.

**Rating:**

Needs Clarification (NC): a submission which does not meet the reviewing criteria and needs clarification for its described problem or solution

**Strengths And Weaknesses:**

Strengths:
* The paper is well-organized, with clear headings and subheadings, making it easy for readers to follow the methodology and results. From a reproducibility perspective, it is noted that authors will provide code and data information upon acceptance.
* The study uses multiple sources of bibliographic data, including IEEE Xplore, Springer, and Scopus, to provide a comprehensive analysis of AI in colonoscopy research.
* The methodology used to generate topic clusters - non-negative matrix factorization (NMF) models is a useful method for organizing and understanding large amounts of text data via topics and its associated keywords.

Weaknesses:
* While the paper's abstract provides a comprehensive overview of the research, the introduction falls short in describing the general research conducted on colonoscopy and the previous studies analyzing AI for colonoscopy research via scholarly publications. It would be beneficial to briefly discuss this information in the introduction to provide context for the reader. Currently, the introduction only describes the Litstudy Python library, but including additional background information would make the chosen topic more understandable and relevant to the reader.
* The paper's methodology section is well-described, outlining the steps followed in the data pre-processing and analysis stages. However, although broad statistics are calculated about the document's metadata, including per author and per journal, the paper only presents information about publications per year while excluding other statistics that were listed. Additionally, there is no detailed explanation provided for the threshold value of 0.85 used in the NMF model and how it was chosen, which is crucial in understanding the accuracy and reliability of the generated topic clusters. A more thorough explanation of these aspects would enhance the reader's comprehension of the methodology and results.
* Moreover, on the last part of methodology, it would be nice to modify the Figure 2 network visualization to better illustrate the connection between the publications.
* In the results sections, naming the topics with its associated topic keywords would help the reader to understand the content better. Additionally, the paper claims that Topic 10 is a weak relationship among topic clusters, providing evidence for the claim with explaining statistics associated with topics such as probability, document similarity or cross entropy score would be helpful.
* The paper provides supporting figures, however, some of the figure titles don't match the figure content and providing detailed titles would support the reader understanding of the result.
* In addition to conclusion, adding a discussion of the limitations of the study, such as the exclusion of certain databases or the potential bias in the selection of keywords. Addressing these limitations would allow other researchers for further exploration of the topic.


**Suggested Changes:**

To strengthen the paper, it would be beneficial to solidify the importance of the research questions and provide a clear explanation of previous research and its shortcomings, as well as the benefits of the current study. Additionally, explaining the Python libraries used to conduct the co-citation network analysis and topic modeling with NMF in detail would help other researchers to reproduce the methods and continue this study.

Although the paper's results are mainly explained in a plot format, utilizing graph network analysis would provide a better explanation and illustration of the connections between publications, authors, and topics. Furthermore, for the topic modeling part of the results, adding a probability and similarity score associated with each topic and document would assist in understanding the importance of each topic. Incorporating these suggestions would enhance the paper's comprehensiveness and usefulness for future research.

---

### Official Review · Reviewer_5hDi · 2023-04-01

**Confidence:** 5

**Summary Of Contributions:**

The paper presents a study that aims to explore the past work on the use of artificial intelligence (AI) in colonoscopy.  The study utilized various methods, including bibliographic network analysis, topic modelling using natural language processing (NLP), and co-citation analysis.

**Rating:**

Great Start (GS): a submission which meets some of the reviewing criteria but has room for improvement

**Strengths And Weaknesses:**

STRENGTHS
1. The use of bibliographic network analysis and topic modelling using NLP allowed the authors to identify topics and their relationships, providing a comprehensive view of the research area.
2. The author made use of co-citation analysis, which provides insight into document similarity and can be used to identify potential future research areas

WEAKNESS
1. The author only used papers retrieved from Scopus, which may not be representative of all relevant research on AI in colonoscopy.
2. The study does not provide a detailed explanation of the threshold used for topic modelling or the specific algorithms used for co-citation analysis, which may limit the reproducibility of the study.

**Suggested Changes:**

1. To address the weakness of the study, the authors could consider including papers from additional databases to ensure that the findings are representative of all relevant research.

2. Providing more details about the methodology, including the specific algorithms used for co-citation analysis and the rationale for the threshold used for topic modelling, would enhance the reproducibility of the study.

---

> ### Author Response · Authors · 2023-05-31
> **Revision**
>
> 1.)The research primarily focused on AI and colonoscopy. We planned to include more keywords for a broader review using another method.

---

### Official Review · Reviewer_1SGs · 2023-04-02

**Confidence:** 3

**Summary Of Contributions:**

Paper involves creating topic clusters on scholarly publications of artificial intelligence (AI) for colonoscopy.Authors have used Litstudy library. Using the analysis, authors claim that this study enables the selection of specific scientific articles. It also helps in examination and visualization of their scholarly contribution and reveal insights into word clusters in AI for colonoscopy publications.

**Rating:**

Great Start (GS): a submission which meets some of the reviewing criteria but has room for improvement

**Strengths And Weaknesses:**

Strengths:

- Findings were communicated clearly and effectively.
- Claims and conclusions are partially justified by the findings.
- Topic modeling part can be reproduced.
- Format looks fine.

Weaknesses

- Creating topic clusters is a common technique. More relevant literature can be added where similar studies have been done - may be the ones that used similar dataset if not exactly the same.
- Claims and conclusions are partially justified by the findings. Artificial intelligence might not be the only word that signifies the use of artificial intelligence. There are word other than artificial intelligence like machine learning, inference etc. Also the dataset used consists of only papers from IEEE Xplore and Springer. There are so many other avenues where papers are published that might involve usage of machine learning in this field.
- Notebook and data would be added after the paper is accepted. Topic modeling part can be reproduced. But it is not clear how cocitation graphical network was created in Figure 2.
- Format looks fine.

**Suggested Changes:**

- It is mentioned in the abstract that topic 5 with AI was the vaguest word in the topic clouds. It would good to add what vague means here.
- Secondary data generated were topic clusters from abstract and bibliographic cocitation networks. It would be good to add few example data points. Everyone understands abstract but a bibliographic cocitation network is not known to everyone.
- In section 2.2, it would be good if we can add block digram that describes the transformation pipeline end to end.
- Bigger Figure 2 would lead to better understanding of statements in results section.
- It is difficult observe “patient” in topic 3 in figure 1.

---

> ### Author Response · Authors · 2023-05-31
> **Revision as advised**
>
> 1.) Added a review on topic clusters, which used a similar method for bibliographic citation analysis  2.) Replaced the word 'vaguest' with the lowest topic phrase weight. 3.) Added a cocitation graph review with author Song(2023).4.) Added a larger version of figures 1 and 2 in the appendices.5. Added framework by the owner of Litstudy as shown in the appendices.

---

### Comment · Program_Chairs · 2023-06-15
**Author email format needs correction**

Dear authors,

In the de-anonymized camera-ready version of your submission, the email format is confusing and wrong. Commas are used to separate two email addresses from the same domain, and curly brackets are used to group them.

For example
{firstname.lastname, first-name.last-name}@domain

The title prefixing a person's name should also be omitted.

Please revise.

-PCs

---

### Meta-Review · Area_Chair_d7T6 · 2023-04-02

**Recommendation:** Invite to revise
**Confidence:** 5

**Metareview:**

On the strength side, this work analyzed bibliographic networks, topics and co-citations of the concerned publications, with bibliographic data from multiple sources. However, the writing of this paper needs much improvement:

- The abstract contains too many implementation details rather than a brief overview of the paper.
- There lacks a description on the background of research and motivations.
- Results and the figures are not clearly presented.

More importantly, this paper focuses too much on technical and implementation details, and it only has a specific case study on publications about AI for colonoscopy. There are no methodology contributions or findings that are generally applicable beyond the case study.


**Summary:**

This work analyzes existing publications on AI for colonoscopy.

**Reason For Not Giving A Higher Recommendation:**

The writing needs significant improvement. And the paper is only about a narrow case study, with no methodology contributions or findings that are generally applicable beyond the case study.


**Reason For Not Giving A Lower Recommendation:**

N/A

---

### Decision · Program_Chairs · 2023-04-08

Revision not accepted; not invited to archive